# Multimodal Causal Reasoning for UAV Object Detection

**Nianxin Li[1], Mao Ye[1],\* Lihua Zhou[2], Shuaifeng Li[1], Song Tang[3], Luping Ji[1], Ce Zhu[1]**
[1]University of Electronic Science and Technology of China
[2]CAIR, HKSIS, CAS
[3]University of Shanghai for Science and Technology, China
linianxin1220@gmail.com, cvlab.uestc@gmail.com
https://github.com/lnxwow/MCR-UOD

## Abstract

Unmanned Aerial Vehicle (UAV) object detection faces significant challenges due to complex environmental conditions and different imaging conditions. These factors introduce significant changes in scale and appearance, particularly for small objects that occupy limited pixels and exhibit limited information, complicating detection tasks. To address these challenges, we propose a Multimodel Causal Reasoning framework based on YOLO backbone for UAV Object Detection (MCR-UOD). The key idea is to use the backdoor adjustment to discover the condition-invariant object representation for easy detection. Specifically, the YOLO backbone is first adjusted to incorporate the pre-trained vision-language model. The original category labels are replaced with semantic text prompts, and the detection head is replaced with text-image contrastive learning. Based on this backbone, our method consists of two parts. The first part, named language guided region exploration, discovers the regions with high probability of object existence using text embeddings based on vision-language model such as CLIP. Another part is the backdoor adjustment casual reasoning module, which constructs a confounder dictionary tailored to different imaging conditions to capture global image semantics and derives a prior probability distribution of shooting conditions. During causal inference, we use the confounder dictionary and the prior to intervene on local instance features, disentangling condition variations, and obtaining condition-invariant representations. Experimental results on several public datasets confirm the state-of-the-art performance of our approach. The code, data and models will be released upon publication of this paper.

## 1 Introduction

Deep learning has driven remarkable progress in object detection, with models such as YOLO[15] and Faster-RCNN[11] achieving strong performance on standard datasets. However, these methods struggle with Unmanned Aerial Vehicle (UAV) imagery due to unique aerial imaging challenges. The bird's-eye view introduces dense and cluttered backgrounds, where target objects are often obscured by complex environmental patterns. Combined with varying lighting and weather conditions, these factors create severe interference that disrupts feature learning and localization. Consequently, existing detectors suffer from high false alarm rates and missed detections, limiting their effectiveness in critical UAV applications such as surveillance and disaster monitoring. Developing robust algorithms to overcome these background interference challenges remains an open research problem.

---

\*Corresponding author.

39th Conference on Neural Information Processing Systems (NeurIPS 2025).

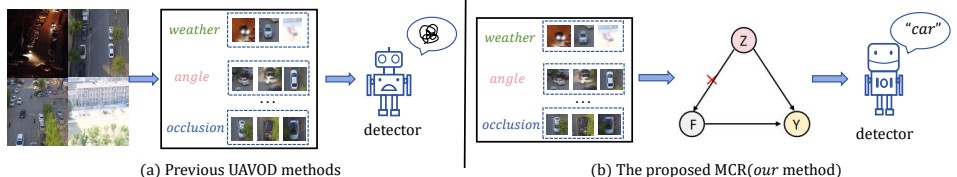

(a) Previous UAVOD methods        (b) The proposed MCR(*our* method)

Figure 1: (a) Previous methods do not handle confounders in UAV images, confusing detector. (b) Our approach removes the confounders via backdoor causal reasoning, enabling a better detector. The previous methods are mainly divided into three routes. The first focuses on region-based strategies, selectively up-scaling regions with dense objects to improve detection accuracy [14, 10, 43, 28]. The second route introduces additional network modules, such as attention mechanisms and multi-scale feature fusion, to enhance feature representations [58, 55, 30, 52]. The final employs data augmentation techniques to increase data diversity, allowing the models to handle a wider range of scenarios [42, 54]. Although these methods have made progress, they still have the following aspects for improvement: 1) Previous works rely solely on single visual features, limiting the improvement of detector performance; 2) Feature enhancement through attention mechanisms lacks interpretability, and does not clearly address how confounders like interference caused by redundant background information or challenging shooting conditions are mitigated.

To address the above issues, we propose the following solutions. First, utilizing a vision-language model for detection enables the full utilization of multimodal knowledge. By integrating the detection network with a vision-language model, such as CLIP [34], the ability of the model to understand and correlate visual and textual information can be improved. This multimodal integration allows for a more context-aware understanding of the objects in the scenarios, improving the detection process, especially in scenarios where visual cues alone cannot offer enough discriminate information. Second, we incorporate causal inference [32, 31], a framework designed to model cause-and-effect relationships, which allows us to systematically address confounding effects [33]. Specifically, we focus on intra-class inconsistencies caused by factors such as environment and shooting conditions. Using causal reasoning, we can effectively eliminate the influence of these confounders, leading to more robust and accurate object detection, even in challenging environments. Finally, relying on visual-language models pre-trained on large data, we can construct confounding factors from a textual perspective that are difficult to capture from an image acquisition perspective. In this way, the detection model that learns to remove these confounding factors has a stronger and more robust generalization ability.

Based on the above analysis, we propose a novel method called Multimodal Causal Reasoning for UAV Object Detection (MCR-UOD). Specifically, this method consists of two modules. The first is the language-guided region exploration (LGRE) module. This module leverages the synergy between visual and linguistic information to guide the detection process. By integrating multimodal knowledge, LGRE endows the model with the ability to focus on object regions, improving detection accuracy in complex scenarios where visual information is insufficient. It first encodes category names to features using the CLIP text encoder and then computes the object existence score for each region in the visual feature map based on these text features, highlighting regions more likely to contain objects. This process helps to remove the intervention from the background. The second is the Backdoor Adjustment Causal Reasoning (BACR) module, which aims to construct a confounder dictionary based on CLIP text embeddings to eliminate the interference of confounders on the objects, enabling the model to handle intra-class feature inconsistencies due to environment and shooting condition variations. This module first selects regions based on the object existence scores. Then it applies a do-operator using the confounder dictionary to enhance the selected features and updates the confounder dictionary accordingly.

Our contributions can be summarized as follows: (1) We are the first to apply the mathematical principles of causal inference to UAV object detection. The effects of varying confounders, caused by environment and shooting conditions, are weakened, allowing the detectors to ensure feature consistency within the same category, thus enhancing the robustness and accuracy of object detection. (2) Unlike previous causal inference methods in computer vision problems, we utilize text embeddings to construct and initialize a confounder dictionary, achieving cross-modal deconfounding. Due to the diversity of UAV environments and shooting conditions, it is clearly impossible to rely on UAV images to construct a confounder dictionary. The vision-language model provides the possibility to construct such a dictionary. (3) Unlike traditional methods that rely solely on visual features, we

propose a novel approach that leverages multimodal knowledge for UAV object detection. Based on text embeddings, the detector backbone will focus more on the object regions. So, although our method is based on the single-stage YOLO backbone, we can still obtain the region of interest and apply causal reasoning to enhance its features.

## 2   Related Work

**UAV object detection (UAVOD)** refers to the task of identifying and localizing objects in UAV-captured images. The existing approaches can be categorized into three main routes. The first is zoom-in strategies, which improve detection accuracy by selectively upscaling regions containing dense or small objects. For example, EVORL employs an evolutionary reinforcement learning mechanism guided by a reward function to determine optimal image patch scales [51], while AdaZoom dynamically adjusts the size and aspect ratio of zoomed regions according to the spatial distribution of small objects [43]. The second route enhances feature representation through additional network modules, such as attention mechanisms or multi-scale fusion. The representative method TPH-YOLOv5 integrates the Transformer and CBAM (Convolutional block Attention Module) to emphasize key regions and improve feature extraction capability [58]. Other works introduce lightweight modules or hierarchical attention to further improve performance [52, 30]. The third route adopts data augmentation techniques to increase sample diversity and improve generalization in different scenarios. These methods generate varied input distributions through weather simulation, viewpoint changes, or domain transfer to train better detectors [42, 54].

**Vision-language model for object detection approach** generally falls into two categories. The first focus on mapping language representations to region prompts. Coop optimizes class prompts with contrastive learning to boost the performance of the vision-language model [57]. GLIP treats object detection as an association problem, aligning regions or bounding boxes with corresponding text prompts [18]. The second category integrates the vision-language model into the existing detection framework. DenseCLIP adapts CLIP for dense prediction by integrating feature adaptation and dense semantic guidance to enhance localization and segmentation tasks [35]. ProposalCLIP employs an unsupervised approach to directly label images of objects [36], while RegionCLIP applies region-based pre-training to associate image regions with textual descriptions [56]. YOLO-World uses region-text alignment with CLIP features to enhance open-vocabulary detection, which employs multi-scale feature fusion and query-based decoding to improve generalization to unseen objects [4].

**Causal inference** is increasingly applied to visual tasks, with methods generally categorized as explicit confounder construction and implicit intervention removal [49, 25]. The former approach builds dictionaries based on object features or relationships and applies do-operation to remove confounding effects [41, 50, 29, 17, 45]. VC R-CNN computes the frequency of occurrence of each category as a prior probability and leverages prototypes to mitigate the negative impact of confounder objects during relational reasoning [41]. MAWCA constructs and updates a confounder dictionary using ROI features, then uses a Transformer during inference to obtain interference-free object features, allowing transfer across different weather conditions [50]. The latter estimates causal effects by using sample augmentation or attention mechanisms to infer results under different interventions and averaging them. CT-MRI achieves single-source domain generation by randomly applying various style augmentations to regions in the image, thereby obtaining domain-invariant features [45]. CIRL learns causal representations that can mimic causal factors based on the ideal properties emphasized, thereby enhancing the robustness of the learned features [27].

## 3   Preliminaries

**Problem statement.** Suppose that the UAV object detection training set $D_{train} = \{X_{tr}^i, Y_{tr}^i\}_{i=1}^{N_{tr}}$, where $Y_{tr}^i = (b_{tr}^i, c_{tr}^i)$ represents the boxes and classes of objects in the $i$-th training image. $N_{tr}$ is the cardinality of the training set. The test set is $D_{test} = \{X_{te}^i, Y_{te}^i\}_{i=1}^{N_{te}}$, where $N_{te}$ is the cardinality of the test set and $Y_{te}$ is unknown. Our goal is to use a vision-language model (CLIP) and casual reasoning to train a better object detector, improving detection performance on UAV images.

**Structural causal model.** As illustrated in Fig.2, we construct a structural causal model to represent the relationships between variables in the detection process. $F$ denote the features of the objects, $Z$ represent the confounders, and $Y$ are the classification results of the objects; the directed edges

represent the causal relationships between the variables. In the directed acyclic graph, the path $F \rightarrow Y$ indicates that the features of objects $F$ directly influence their classification results $Y$. For instance, distinctive visual patterns (e.g. wheels for cars) causally determine the predictions of the model. The backdoor path $F \leftarrow Z \rightarrow Y$ highlights the confounding effect of $Z$, which introduces spurious associations between features $F$ and predictions $Y$. For example, $Z$ could represent environmental factors (e.g. lighting, weather), data set biases (e.g. class imbalance), or sensor distortions (e.g. camera noise). These confounders corrupt the feature representation process ($F \leftarrow Z$) while simultaneously influencing the model's decision ($Z \rightarrow Y$). For example, poor lighting ($Z$) may degrade visual features ($F$), making objects harder to recognize, while also skewing label distributions ($Y$) if certain classes dominate low-light scenarios. Such backdoor paths can lead models to rely on non-causal shortcuts. To address this, we employ a backdoor adjustment to isolate the genuine causal relationship $F \rightarrow Y$.

**Backdoor adjustment for causal learning.** To eliminate the confounding bias introduced by $Z$, we implement a backdoor adjustment method based on causal inference theory. The core idea is to stratify the data according to the confounder $Z$ and then calculate the weighted average of the predictions in all strata. Formally, the causal effect of $F$ on $Y$ can be estimated as

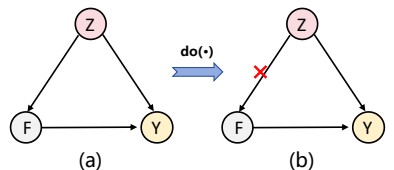

$$P(Y|do(F)) = \sum_z P(Y|F, Z=z)P(Z=z), \quad (1)$$

where the *do*-operator signifies an intervention that removes the influence of $Z$. Here, $P(Y|F, Z=z)$ represents the prediction conditioned on both features $F$ and a specific value of $Z$, while $P(Z=z)$ accounts for the prior distribution of the confounder. In practice, we first

Figure 2: (a) Structural causal model shows the direct effect $F \rightarrow Y$ and confounding path $F \leftarrow Z \rightarrow Y$; (b) Intervention model where the confounding path $F \leftarrow Z$ is blocked (indicated by ×), enables estimation of the true causal effect $F \rightarrow Y$.

discretize continuous confounders (e.g., lighting levels) into interpretable bins with a confounder dictionary $Z$. Then, we train our model to estimate $P(Y|F, Z)$ and compute $P(Z)$ empirically from the training data. Finally, we aggregated the predictions across all $Z$ strata to obtain an unbiased estimate of $P(Y|do(F))$. This approach effectively blocks the backdoor path $F \leftarrow Z \rightarrow Y$, ensuring that the learned relationships reflect true causal mechanisms rather than spurious correlations. For further technical details, see Section 4.2.

## 4 Proposed Method

**Overview.** As shown in Fig.3, the proposed method consists of two parts: the *Language Guided Region Exploration (LGRE)* module and the *Backdoor Adjustment Causal Reasoning (BACR)* module. The first module selects possible regions where objects may exist, and then the second module refines the features by backdoor adjustment ignoring the effects of confounders. Specifically, for the UAV image $X_i$, multilevel features $\{C_1, C_2, C_3\}$ can be obtained based on YOLOv8 [40]. Since the challenges of UAVOD stem primarily from small objects, we focus on refining the $C_1$ layer features. In the *LGRE* module, $K$ category prompts and the corresponding text embeddings $\{e_k\}_{k=1}^K$ are obtained using the CLIP text encoder [39]. $K$ is also the number of categories. Then, the object existence probability map $s$ for each pixel is obtained by taking the product of the text embeddings $\{e\}_{k=1}^K$ and the low-level feature $C_1$. In the *BACR* module, the GPT model [1] is used to generate a series of confounder prompts, which are encoded as text embeddings $Z$ to construct and initialize a confounder dictionary. Based on the probability map $s$, we perform causal interventions on the features with high probability. The modulated features $F'$ are projected back into the original feature map $C_1$, resulting in a new feature map $C_1^n$. Finally, we input $\{C_1^n, C_2, C_3\}$ along with the text embedding $E$ into the contrastive head and the box head to obtain the final detection results.

### 4.1 Language Guided Region Exploration

In UAVOD, one of the key challenges is to accurately localize small objects, which are often represented in low-level features. To address this issue, we leverage the rich prior knowledge learned by language models to enhance object localization. The core idea is to utilize textual information to guide the model's attention towards regions in the image that are likely to contain objects, improving

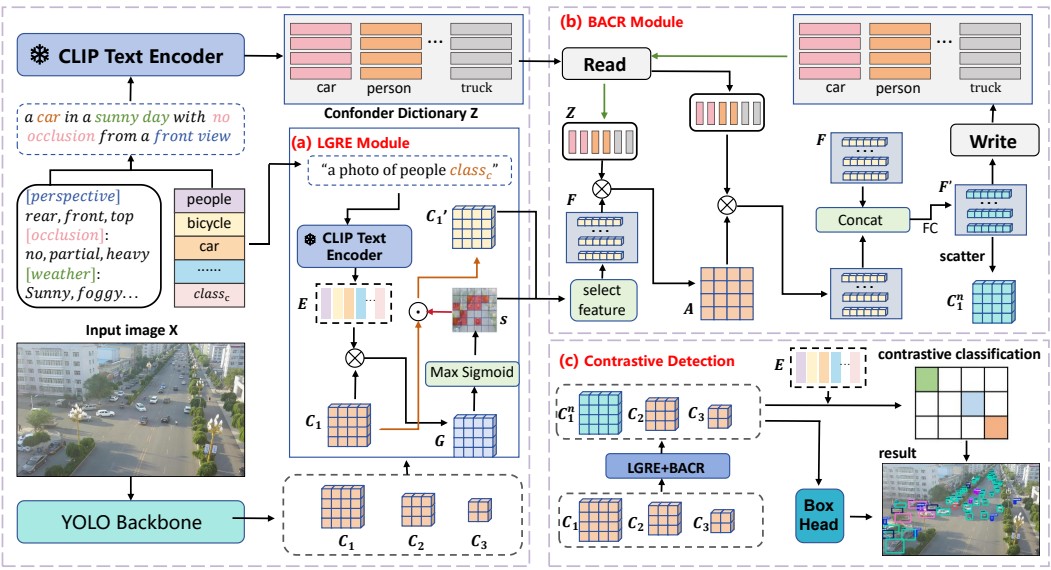

Figure 3: The overall framework of MCR-UOD. (a) Language Guided Region Exploration (LGRE) module computes object existence probability map using CLIP text embeddings and (b) Backdoor Adjustment Causal Reasoning (BACR) module performs causal intervention through cross-attention between selected high-probability pixel-level visual features and confounder dictionary. The deconfounder feature map $C_1^n$ is then fed into detection head.

the object existence score for different regions of interest. Specifically, we generate $K$ prompts by constructing descriptive text based on category names, such as 'a photo of [category]', and obtain text embeddings $E = \{e_k\}_{k=1}^{K}$ by encoding these prompts using the CLIP text encoder [39], where $E \in \mathbb{R}^{K \times D_1}$, and $D_1$ is the dimension of the text embedding. The input image $X$ is encoded by CSPDarknet [2], generating multi-scale feature maps $\{C_1, C_2, C_3\}$. Since the challenges of UAVOD are primarily stemmed from small objects in low-level features, we focus on $C_1 \in R^{H \times W \times D_2}$ accordingly, where $H$, $W$ and $D_2$ denote height, width, and channel number, respectively. First, the similarities between pixel-level features in $C_1$ and text embeddings $\{e_k\}_{k=1}^{K}$ are calculated to obtain the score map $G$ as follows,

$$G = f_1(C_1) \times (f_2(E))^T, \quad G \in R^{H \times W \times K} \tag{2}$$

where $f_1$ and $f_2$ represent two fully connected layers, which transform $C_1$ and $E$ into the same dimension feature space, respectively. Then we calculate the maximum value of $G$ at each pixel and apply Sigmoid operation as the following,

$$s(h, w) = \sigma \left( \max_k G(h, w, k) \right), \quad s \in R^{H \times W \times 1} \tag{3}$$

where $(h, w)$ represents the pixel coordinate, and $s$ indicates the likelihood of object existence at this pixel. According to the object existence probability map $s$, the regions that are likely to contain objects can be selected. By perform element-wise multiplication with $C_1$ and $s$, the original feature map $C_1$ can be updated as

$$C_1' = C_1 \odot s, \quad C_1' \in R^{H \times W \times D_2} \tag{4}$$

where $\odot$ represents the element-wise multiplication, and $C_1'$ is the updated $C_1$ feature map.

## 4.2 Backdoor Adjustment Causal Reasoning

In LGRE module, the updated low-level features $C_1'$ and the corresponding object existence probability map $s$ are obtained. We select the highest $\tau$ pixel features to form a feature matrix $F \in R^{N \times D_2}$ based on the probability values as follows,

$$F = \{C_1'(i) | i \in I_\tau\}, \quad I_\tau = Top_N(s), \quad N = \tau \cdot HW \tag{5}$$

where $\tau$ is a threshold hyperparameter, $N$ is the number of selected pixels. This feature matrix $F$ records the regions that are more likely to contain objects. According to Eq.(1) in Section 3, we implement backdoor adjustment for visual features $F$ using a confounder dictionary $Z$. $P(Y \mid F, Z = z)$ refers to the classification result $Y$ obtained from the features $F$, given $Z = z$. The features $F$ first pass through a function $f$ with cross-attention to the dictionary $Z$ to obtain the classification logits, which are then passed through a softmax operation to produce the classification result $Y$. Therefore:

$$P(Y \mid do(F)) = \sum_z P(Y|F, Z = z)P(Z = z) = \mathbb{E}_z[Softmax(f(F, z))]. \tag{6}$$

We utilize the NWGM [44] method to approximate the aforementioned expectation. In brief, NWGM efficiently transforms the outer expectation into the Softmax function as follows:

$$\mathbb{E}_z[Softmax(f(F, z))] \overset{\text{NWGM}}{\approx} Softmax(\mathbb{E}_z[f(F, z)]). \tag{7}$$

Thus, the causal intervention is to compute the object probability under different confounder conditions $z \in Z$, blocking the confounding path $F \leftarrow Z \rightarrow Y$ through expectation marginalization.

**Confounder dictionary construction.** It is extremely difficult to collect confounder images for different UAV shoot conditions and scenarios. However, by fully utilizing multi-modal knowledge, we can construct and initialize the confounder dictionary through text prompts. Specifically, we employ the language model GPT [1] to generate texts of different confounders, such as "a photo of a car in a rainy day with no occlusion from a rear view". The confounders include weather (sunny, rainy, foggy, nighttime), occlusion (no, partial, heavy), and perspective (front, side, rear, top). In this way, we obtain $K \times M$ prompts $\{PT_m\}_{m=1}^{K \times M}$, where $M$ is the number of combinations of confounders. The corresponding text embeddings $Z \in R^{S \times D_2}$ are obtained using the CLIP text encoder [39]. These embeddings initialize the confounder dictionary, where $S = K \times M$ represents the number of items in the dictionary. To facilitate subsequent operations, we unify the dimensions of text and image representations as $D_2$.

**Causal reasoning.** Cross-attention is performed to complete the task in Eq.(6). The selected features $F$ are projected as query embeddings $Q \in R^{N \times D_2}$ via a linear mapping, and the confounder dictionary $Z$ is independently projected as key and value embeddings $K \in R^{S \times D_2}$ and $V \in R^{S \times D_2}$, respectively. Formally,

$$Q = W_q * F + b_{query}, K = W_k * Z + b_{key}, V = W_v * Z + b_{value}, \tag{8}$$

where $W_q, W_k$, and $W_v$ are the parameters of the linear transformation layers. $b_{query}, b_{key}$, and $b_{value}$ are the corresponding bias values, respectively. Then we calculate the attention weight matrix $A$ through dot product operation and apply Softmax function for normalization as

$$A = softmax(\frac{QK^T}{\sqrt{D_2}}), \tag{9}$$

$A \in R^{N \times S}$ assigns soft weights to the probability of each confounder interfering with the feature $F$, which allows us to approximate the expectation over confounders in the causal prediction formulation Eq.(7). Specifically, we model $\mathbb{E}_z[f(F, z)]$ by fusing the original feature $F$ with a weighted confounder context $AV^\top$. The combined representation is then transformed via a learnable function $f_F$ to produce a refined feature:

$$F' = f_F(Cat(F, AV^T)), \quad F' \in \mathbb{R}^{N \times D_2}. \tag{10}$$

This enables the network to learn confounder-aware representations while preserving task-relevant information. $f_F \in \mathbb{R}^{2D_2 \times D_2}$ is a fully connected layer, and $Cat()$ denotes the concatenation operation. $F'$ represents the features after applying the do-operator, corresponding to $\mathbb{E}_z[f(F, z)]$ in Equation 6. Subsequently, $F'$ is passed through the classification head to obtain $Softmax(\mathbb{E}_z[f(F, z)])$. We interpolate $F'$ into the feature map $C_1'$ according to the indices $I$ to obtain the final feature $C_1^n$, which, along with $C_2$ and $C_3$, is fed into the detection head.

**Dictionary update.** To ensure the effectiveness and representativeness of the confounder dictionary, we continuously update the items in the dictionary during forward propagation of the network. Specifically, we fuse the visual feature $F_p$ at some pixel into the most similar confounder $z_i$ from the confounder dictionary based on the similarity as follows,

$$z_i^{t+1} = \alpha z_i^t + (1 - \alpha)F_p \tag{11}$$

Table 1: Comparison of different approaches on UAVDT and VisDrone. The best and second best values are highlighted in bold and red, respectively.

| Method | Backbone | UAVDT | | | VisDrone | | |
|---|---|---|---|---|---|---|---|
| | | $AP$ | $AP50$ | $AP75$ | $AP$ | $AP50$ | $AP75$ |
| FPN [22] (NeurIPS,2015) | ResNet-50 | 16.9 | 30.7 | 17.2 | 16.9 | 30.7 | 17.2 |
| Faster R-CNN (TPAMI,2017) | ResNet-50 | 12.1 | 23.5 | 10.8 | 21.8 | 41.8 | 20.1 |
| CascadeRCNN [3] (CVPR,2018) | ResNet-50 | 17.1 | 30.5 | 18.6 | 23.6 | 38.9 | 24.6 |
| ClusDet [46] (ICCV,2019) | ResNet-101 | 13.7 | 26.5 | 12.5 | 32.4 | 56.2 | 31.6 |
| DMNet [20] (CVPR,2020) | ResNet-50 | 14.7 | 24.6 | 16.3 | 28.2 | 47.6 | 28.9 |
| GLSAN [5] (TIP,2021) | ResNet-50 | 17.0 | 28.1 | 18.8 | 30.7 | 55.4 | 30.0 |
| AdaZoom [43] (TMM,2022) | ResNet-50 | 20.1 | 34.5 | 21.5 | 40.3 | 66.9 | 41.8 |
| Zoom&Reasoning [10](SPL,2022) | ResNet-50 | 21.8 | 34.9 | 24.8 | 39.0 | 66.5 | 39.7 |
| UFPMPDet [14] (AAAI,2022) | ResNet-50 | 24.6 | 38.7 | 28.0 | 36.1 | 57.3 | 38.2 |
| EVORL [51] (AAAI,2024) | ResNet-50 | 28.0 | 43.8 | 31.5 | 42.2 | 66.0 | 44.5 |
| TPH-YOLOv5 [58] (ICCV,2021) | CSPDarknet | 26.9 | 41.3 | 32.7 | 42.1 | 63.1 | 45.7 |
| TPH-YOLOv5+ [55] (MDPI,2023) | CSPDarknet | 30.1 | 43.5 | 34.3x | 41.4 | 61.9 | 45.0 |
| UAV-YOLOv8 [40] (MDPI,2023) | CSPDarknet | 27.3 | 42.1 | 30.4 | 42.7 | 65.5 | 44.7 |
| SPAR [19] (AAAI,2025) | CSPDarknet | 30.5 | 43.9 | 34.7 | 42.8 | 66.7 | 45.1 |
| **MCR-UOD** (ours) | CSPDarknet | **31.4(+1.3)** | **44.7(+0.8)** | **35.6(+0.9)** | **44.6(+1.8)** | **67.3(+0.4)** | **47.5(+1.8)** |

where $\alpha = 0.05$ is a trade-off weighting parameter and $z_i^{t+1}$ is the updated confounder.

**Remark.** Unlike previous methods that rely solely on visual data [41] , we propose a multi-modal approach that constructs the confounder dictionary using language-generated prompts. These prompts are encoded via the CLIP text encoder, enabling explicit and controllable modeling of various confounders. These confounders are difficult to collect on the basis of visual images. Our strategy improves the adaptability of the dictionary and improves causal reasoning in complex aerial scenarios.

### 4.3 Detection Head and Loss Function

Following YOLOv8 [40], we use a decoupled head with two 3×3 convolutions to obtain the object bounding boxes $\{b_j\}_{j=1}^{J}$ and object embedding $v_j$, where $J$ denotes the number of objects. Additionally, we replace the original classification head with a text contrastive head. The class probability vector $c$ is computed as follows,

$$c = \alpha \cdot \frac{v_j}{||v_j||} \cdot \frac{E^T}{||E||} + \beta \tag{12}$$

where $v_j$ is the object embedding, $E$ is the category text embeddings, $||\cdot||$ is the $L_2$ norm. In addition, we add an affine transformation, where $\alpha$ is the learnable scaling factor and $\beta$ is the learnable shifting factor. Our method follows the same end-to-end training approach as YOLOv8 [40]. Additionally, to better handle the large number of small objects in UAV images, we replace the original IoU loss with the more effective WIoU loss [37] as

$$\mathcal{L}_{WIoU} = 1 - e^{-r(h^* w^*)} \cdot \frac{|B \cap B^*|}{|B \cup B^*|} \tag{13}$$

where $B$ and $B^*$ represent the predicted bounding box and the ground truth bounding box, respectively. $r = 0.05$ is a fixed hyperparameter that controls the decay rate of the weight factor; $h^*$ and $w^*$ are the height and width of $B^*$, respectively. The overall training loss is

$$\mathcal{L} = \mathcal{L}_{\text{WIoU}} + \mathcal{L}_{\text{DFL}} + \mathcal{L}_{\text{cls}} \tag{14}$$

where $\mathcal{L}_{\text{cls}}$ is the classification loss, always calculated using cross-entropy, and $\mathcal{L}_{\text{DFL}}$ is the Distribution Focal Loss [16], which improves bounding box prediction accuracy by learning a discrete distribution.

**Training and test**. We integrate the proposed method into the baseline and perform end-to-end training. During both training and inference, the parameters of the CLIP text encoder are frozen, while the confounder dictionary $Z$ is continuously updated.

## 5 Experiments

**Experiment setup.** Three public datasets are used for aerial image object detection: VisDrone [8], UAVDT [7] and HRSC2016 [26]. VisDrone contains 8599 drone-captured images (2000×1500 pixels), split into 6471 for training, 548 for validation, and 1580 for testing. It includes 10 object

| Table 2: Comparison on HRSC2016. | | | |
|---|---|---|---|
| Method | AP | Method | AP |
| R2PN(GRSL'18) | 70.06 | RRD(CVPR'18) | 84.30 |
| RoIT(CVPR'19) | 86.20 | R$^3$Det(AAAI'21) | 89.26 |
| CSL(ECCV'20) | 89.62 | ReDet(CVPR'22) | 90.46 |
| FSM(TPAMI'25) | 91.60 | **MCR-UOD(ours)** | **92.04** |

| Table 3: Ablation study of MCR-UOD. | | | |
|---|---|---|---|
| **Method** | **AP** | **AP50** | **AP75** |
| YOLOv8 | 42.2 | 64.7 | 44.5 |
| +WIOU | 42.5(+0.3) | 65.2(+0.5) | 44.7(+0.2) |
| +WIOU+LGRE | 43.6(+1.1) | 66.5(+1.3) | 45.9(+1.2) |
| MCR-UOD | **44.6(+1.0)** | **67.3(+0.8)** | **47.5(+1.6)** |

categories, mainly vehicles and pedestrians. UAVDT is designed for object detection and tracking, comprising 24143 training images and 16592 testing images (1024×540 pixels). It features diverse aerial scenes and is widely used for detecting cars, trucks, and buses. HRSC2016 contains high-resolution aerial images focused on ship detection, featuring large-scale variations and complex backgrounds. Object detection models are evaluated using standard metrics [9, 23], including AP (Average Precision), AP50 and AP75. We chose YOLOv8 [40] as the backbone of our method and performed all training and validation on two NVIDIA FeForce RTX 3090 GPUs. The number of training epochs is 75, with a batch size of 4. The initial learning rate $lr_0$ is 0.001; the final learning rate $lr_f$ is 0.01; and the weight decay is 0.0005.

## 5.1 Comparison with State-of-the-art Methods

**Compared methods.** To validate the effectiveness of our method, we compare it with several state-of-the-art approaches proposed in recent years.These methods can be categorized according to the descriptions in the related works section as follows: zoom-in strategy based methods including ClusDet [46], DMNet [20], UFPMP [14], Adazoom [43], Zoom&Reasoning [10]; feature representation enhanced methods such as FPN [22], TPH-YOLOv5 [58], TPH-YOLOv5++ [55], SPAR [19],ReDet [12],RoIT [6],R2PN [53],FSM [24],PRD [21]and data augmentation based method PAOD [13], FSM [24], R$^3$Det [48] and CSL [47] .

**Quantitative comparison.** Table 1 presents a summary of the comparison results between our method and nine state-of-the-art approaches on the UAVDT and VisDrone datasets. In terms of three key evaluation metrics, the proposed method significantly outperforms all models compared. Specifically, on the VisDrone dataset the proposed MCR-UOD method achieves an AP of 44.6, representing a 1.8% improvement over the previous best-performance model SPAR; On the UAVDT dataset, MCR-UOD achieves an AP of 31.4, outperforming TPH-YOLOv5 by 1.3 points. Our method achieves an AP50 of 44.6, exceeding SPAR by 0.8 points, and an AP75 of 35.6, outperforming TPH-YOLOv5++ by 0.9 points. These substantial improvements on three metrics demonstrate that our proposed method excels not only in recognizing object regions but also in achieving accurate localization. This indicates that the model is capable of learning more discriminative and fine-grained features, leading to more precise bounding-box regression. Furthermore, on the HRSC2016 dataset, as shown in Table 2, MCR-UOD achieves an mAP of 91.13, outperforming all existing methods. All gains come from the LGRE module for precise region attention and the backdoor adjustment for removing confounding factors, which enhances feature robustness in complex scenes.

**Visualization comparisons.** Fig. 4 provides a comprehensive visualization of the performance of the MCR-UOD method on UAVDT and VisDrone, column (a) shows the original images, while columns (b) and (c) present the detection results of previous state-of-the-art methods, UFPMP and SPAR, respectively. Column (d) illustrates the detection results of our proposed method, MCR-UOD. The regions marked with red circles indicate areas where the previous methods failed to detect objects. In the first row, a truck in a very dark lighting condition is completely missed by the previous methods, but MCR-UOD successfully detects it. In the second row, due to overexposure in the image, the car is difficult to detect using previous methods, while our method correctly identifies it. In the third row, UFPMP and SPAR miss several small objects due to heavy occlusion and their tiny sizes. In contrast, MCR-UOD successfully identifies these small targets. These improvements can be attributed to the core design of MCR-UOD, by integrating vision-language models and causal inference, the method effectively utilizes contextual knowledge and systematically eliminates confounding factors such as lighting and viewpoint changes.

## 5.2 Further Studies

**Ablation study.** Ablation experiments are conducted on VisDrone dataset, as shown in Table 3. Yolov8 [40] is the baseline model; "+WIOU" is the baseline with WIoU loss; "+WIoU+LGRE" denotes using WIOU loss and LGRE module; "MCR-UOD" denotes the complete method. From Table

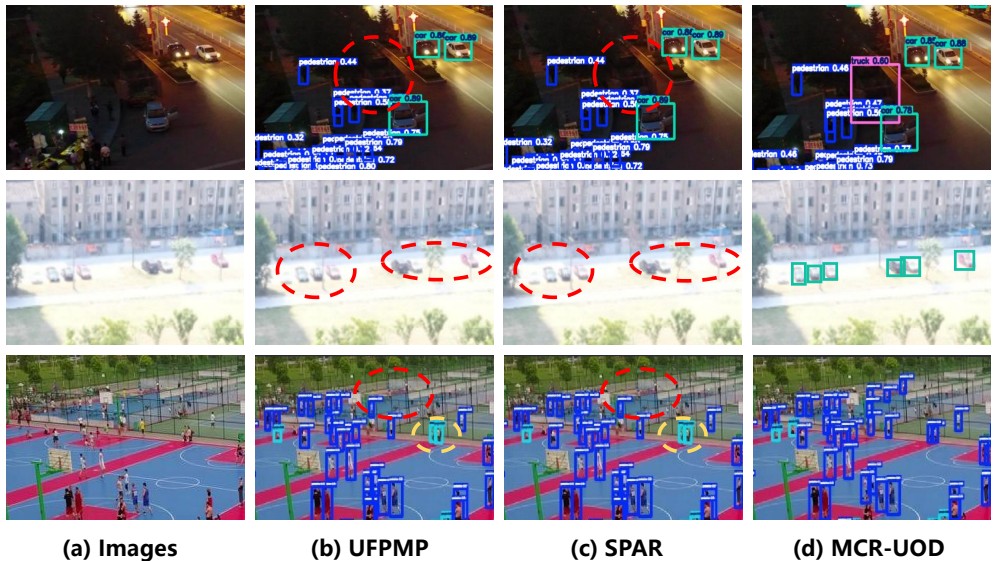

| (a) Images | (b) UFPMP | (c) SPAR | (d) MCR-UOD |

Figure 4: Visualization comparison. (a) shows the original image, while (b), (c), and (d) present the detection results of UFPMP, SPAR, and our proposed MCR-UOD method, respectively. The red circles highlight the objects missed by previous state-of-the-art methods but successfully detected by MCR-UOD. The yellow circles indicate false detections. *Zoom in for best view.*

Table 4: Performance and efficiency comparison across different YOLO backbones.

| Method | AP | AP50 | AP75 | Parameters | GFLOPs |
|---|---|---|---|---|---|
| YOLOv8s | 38.2 | 56.9 | 41.3 | 11.14M | 28.7 |
| YOLOv8m | 40.7 | 59.6 | 42.5 | 25.8M | 79.1 |
| YOLOv8l | 41.1 | 62.8 | 44.0 | 43.6M | 165.4 |
| YOLOv8x | 42.2 | 64.7 | 44.5 | 68.2M | 258.2 |
| YOLOv8s+MCR-UOD | 40.1 | 60.3 | 44.9 | 10.6M | 28.4 |
| YOLOv8m+MCR-UOD | 21.8 | 41.8 | 20.1 | 23.7M | 77.6 |
| YOLOv8l+MCR-UOD | 43.2 | 66.4 | 45.9 | 39.5M | 159.2 |
| YOLOv8x+MCR-UOD | 44.6 | 67.3 | 47.5 | 61.5M | 247.6 |

3, it can be concluded that all modules contribute positively to the final performance. Specifically, the combined use of LGRE and BACR modules results in significantly improved performance compared to baseline; the performance values in terms of AP, AP50 and AP75 in VisDrone are all increased.

**Computational efficiency.** In integrating our method with the backbone, we modify and remove parts of the C2f module, replace the detection head with a text contrastive detector, achieve a lightweight design. Please refer to the appendix for details. Experiments are conducted across YOLOv8 variants (n/s/m/l/x) on VisDrone dataset to explore the speed-accuracy trade-off under different backbone scales. As shown in Table 4, we present the speed and accuracy comparison between models of different sizes. From the table, it is evident that compared to the YOLOv8 baseline, the proposed MCR-UOD achieves faster inference speed and higher detection accuracy across various backbone scales, demonstrating a better trade-off between performance and efficiency. More experimental results are shown in the appendix.

**t-SNE visualization of the BACR module.** To verify the effectiveness of the BACR module, we visualize category-wise features on the VisDrone dataset using t-SNE [38], as shown in Fig. 5. Without BACR, features of the same category are scattered and easily confused due to diverse UAV imaging conditions. In contrast, with BACR, intra-class features become more compact and inter-class boundaries clearer, demonstrating improved feature discrimination for UAV object detection.

**Sensitive analysis.** In our method, there are not many parameters. The only adjustable parameter is $\tau$ in Eq.(5). To investigate the impact of different values $\tau$ on detection performance, we conducted a parameter sensitivity analysis as shown in Figure 6. We systematically tested $\tau$ values within the

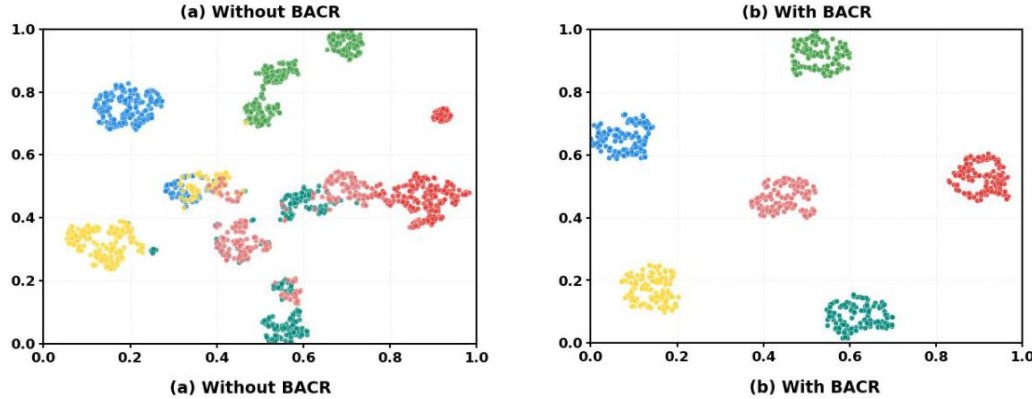

Figure 5: Visualization of t-SNE with and without BACR module.

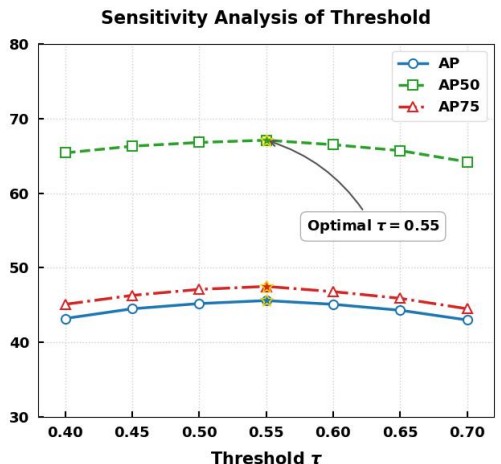

Figure 6: Sensitivity analysis of threshold $\tau$.

range [0.4, 0.7]. The results show that both excessively small and large $\tau$ values lead to reduced accuracy. Overly small $\tau$ values introduce more false negatives by including inaccurate regions, while overly large $\tau$ values produce false positives by missing valid detection areas. The optimal performance is achieved at $\tau$=0.55, which is consequently selected for our experiments. At the same time, we can also observe that as the $\tau$ value changes, the performance changes are also flat. It confirms that the value of $\tau$ is not very sensitive to performance.

## 6 Conclusion

We proposed a new UAV object detection method, called MCR-UOD, which improves the performance of UAV object detection through causal inference and multimodal learning. The framework comprises two key modules: LGRE and BACR. LGRE module leverages a pre-trained vision-language model, such as CLIP text embeddings to compute text-guided attention maps for highlighting possible object regions. The BACR module maintains a dynamic confounder dictionary for causal intervention. Due to the inability to obtain diversity UAV images under various environmental and imaging conditions, the confounder dictionary is constructed and initialized with Clip text embeddings. Backdoor adjustment is applied based on this confounder dictionary that can reduce the influence of confounding factors, thus the extracted object features are imaging condition-invariant and more robust. Experimental results demonstrate that the proposed MCR-UOD outperforms existing methods while maintaining computational efficiency.

## Acknowledgments and Disclosure of Funding

This work was supported by the National Natural Science Foundation of China (62276048, 62476169, 62476049).

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

# A  Illustration of Backdoor Adjustment

Backdoor adjustment is a core method of causal inference used to eliminate confounding bias. The key idea is to adjust a set of confounders $Z$ that block all non-causal paths (ack door) between treatment $X$ and outcome $Y$, allowing estimation of the causal effect $X$ on $Y$.

## A.1  Backdoor Adjustment Formula

If the variable set $Z$ satisfies the backdoor criterion, the causal effect of $X$ on $Y$(Average Causal Effect, ACE) can be estimated using:

$$P(Y = y|do(X)) = \sum_z P(Y = y|X = x, Z = z) \cdot P(Z = z) \tag{15}$$

where the Z must satisfy the backdoor criterion: 1) Blocks all backdoor paths: Z must block every path between X and Y that contains an arrow into X. 2) No new bias introduced: Z must not include any descendants of X.

## A.2  Derivation Process.

The backdoor adjustment is derived using causal graph rules and probability theory, with the following key steps: **From intervention to Conditional Probability.** The intervention do(X=x) corresponds to removing all incoming edges to $X$ in the causal graph and fixing $X = x$. In this intervention, the distribution of $Y$ depends only on $X$ and its parents. If $Z$ satisfies the backdoor criterion, the post-intervention distribution can be expressed as:

$$P(Y|do(X = x)) = \sum_z P(Y|X = x, Z = z) \cdot P(Z = z|do(X) = x) \tag{16}$$

Since do(X=x) does not affect $Z$(because $Z$ is not a descendant of $X$), we have $P(Z = z|do(X = x)) = P(Z = z)$,leading to:

$$P(Y|do(X = x)) = \sum_z P(Y|X = x, Z = z) \cdot P(Z = z). \tag{17}$$

## A.3  Confounder Dictionary Construction.

Collecting confounder images in 218 different images and UAV conditions is extremely challenging. To address this, we fully leverage multi-modal knowledge by constructing and initializing the confounder dictionary using textual prompts. Specifically, we employ the large language model GPT [1] to generate descriptive texts for various confounders, such as "a photo of a car on a rainy day without occlusion from a rear view." The confounders include weather conditions (sunny, rainy, foggy, nighttime), occlusion levels (none, partial, heavy), and viewing perspectives (front, side, rear, top). In this way, we systematically generate linguistic priors for confounder modeling, thus providing rich semantic support for downstream tasks, as shown in Table 5.

# B  More Experiment Results

**Evaluation metrics.** To evaluate the detection performance of our proposed enhanced model, we use several metrics: AP, AP50 and AP75 [9, 23]. The following parameters are utilized: TP (true positives), FP (false positives), and FN (false negatives). Intersection over Union (IoU) measures the overlap between the predicted bounding box and the ground truth box. Precision is defined as the ratio of true positive predictions to the total number of detected samples, calculated as follows:

$$\text{Precision} = \frac{TP}{TP + FP} \tag{18}$$

Recall represents the ratio of true positive predictions to the total number of actual positive samples, calculated as:

$$\text{Recall} = \frac{TP}{TP + FN} \tag{19}$$

Table 5: The 36 prompt templates used in our method, each describing a [CLS] token in various UAV imaging conditions including weather, occlusion, scale, and viewpoint.

| # | Prompt Template |
|---|---|
| 1 | a [CLS] in a sunny scene with no occlusion, viewed from the front at a large scale. |
| 2 | a [CLS] in a sunny scene with no occlusion, viewed from the side at a medium scale. |
| 3 | a [CLS] in a sunny scene with no occlusion, viewed from the rear at a small scale. |
| 4 | a [CLS] in a sunny scene with partial occlusion, viewed from the top at a large scale. |
| 5 | a [CLS] in a sunny scene with partial occlusion, viewed from the front at a medium scale. |
| 6 | a [CLS] in a sunny scene with partial occlusion, viewed from the side at a small scale. |
| 7 | a [CLS] in a sunny scene with heavy occlusion, viewed from the rear at a large scale. |
| 8 | a [CLS] in a sunny scene with heavy occlusion, viewed from the top at a medium scale. |
| 9 | a [CLS] in a sunny scene with heavy occlusion, viewed from the front at a small scale. |
| 10 | a [CLS] in a rainy scene with no occlusion, viewed from the side at a large scale. |
| 11 | a [CLS] in a rainy scene with no occlusion, viewed from the rear at a medium scale. |
| 12 | a [CLS] in a rainy scene with no occlusion, viewed from the top at a small scale. |
| 13 | a [CLS] in a rainy scene with partial occlusion, viewed from the front at a large scale. |
| 14 | a [CLS] in a rainy scene with partial occlusion, viewed from the side at a medium scale. |
| 15 | a [CLS] in a rainy scene with partial occlusion, viewed from the rear at a small scale. |
| 16 | a [CLS] in a rainy scene with heavy occlusion, viewed from the top at a large scale. |
| 17 | a [CLS] in a rainy scene with heavy occlusion, viewed from the front at a medium scale. |
| 18 | a [CLS] in a rainy scene with heavy occlusion, viewed from the side at a small scale. |
| 19 | a [CLS] in a foggy scene with no occlusion, viewed from the rear at a large scale. |
| 20 | a [CLS] in a foggy scene with no occlusion, viewed from the top at a medium scale. |
| 21 | a [CLS] in a foggy scene with no occlusion, viewed from the front at a small scale. |
| 22 | a [CLS] in a foggy scene with partial occlusion, viewed from the side at a large scale. |
| 23 | a [CLS] in a foggy scene with partial occlusion, viewed from the rear at a medium scale. |
| 24 | a [CLS] in a foggy scene with partial occlusion, viewed from the top at a small scale. |
| 25 | a [CLS] in a foggy scene with heavy occlusion, viewed from the front at a large scale. |
| 26 | a [CLS] in a foggy scene with heavy occlusion, viewed from the side at a medium scale. |
| 27 | a [CLS] in a foggy scene with heavy occlusion, viewed from the rear at a small scale. |
| 28 | a [CLS] in a night scene with no occlusion, viewed from the top at a large scale. |
| 29 | a [CLS] in a night scene with no occlusion, viewed from the front at a medium scale. |
| 30 | a [CLS] in a night scene with no occlusion, viewed from the side at a small scale. |
| 31 | a [CLS] in a night scene with partial occlusion, viewed from the rear at a large scale. |
| 32 | a [CLS] in a night scene with partial occlusion, viewed from the top at a medium scale. |
| 33 | a [CLS] in a night scene with partial occlusion, viewed from the front at a small scale. |
| 34 | a [CLS] in a night scene with heavy occlusion, viewed from the side at a large scale. |
| 35 | a [CLS] in a night scene with heavy occlusion, viewed from the rear at a medium scale. |
| 36 | a [CLS] in a night scene with heavy occlusion, viewed from the top at a small scale. |

The average precision (AP) is the area under the precision-recall curve, computed by:

$$AP = \int_0^1 \text{Precision}(\text{Recall}) \, d(\text{Recall}) \tag{20}$$

Mean average precision (mAP) is obtained by averaging the AP values across all sample categories to measure the model's performance across all categories:

$$mAP = \frac{1}{N} \sum_{i=1}^{N} AP_i \tag{21}$$

Here, $AP_i$ represents the AP value for category $i$, and $N$ is the number of categories in the training dataset (in this paper, $N = 10$). AP50 denotes the average precision when the IoU threshold is set to 0.5, while AP75 represents the average precision over IoU thresholds to 0.75.

**Confusion matrix.** From Fig. 7, it can be seen that the diagonal region of the confusion matrix for MCR-UOD is darker in color compared to YOLOv8, indicating that our proposed method has improved the model's ability to correctly predict object categories. This improvement is particularly notable when detecting smaller objects, such as bicycles, tricycles, and awning-tricycles, where our method outperforms YOLOv8. Although there are still some missed detections for these smaller objects in complex backgrounds, our method significantly reduces the proportion of objects misclassified as background compared to YOLOv8. Bicycles, tricycles, and awning-tricycles often appear in dense or occluded environments, making detection in complex backgrounds challenging. Our method improves the feature extraction ability and classification mechanisms of the model, leading to better detection performance and reduced missed detection rates for these small objects. Although the percentage of correctly predicted small objects still needs improvement, our method shows

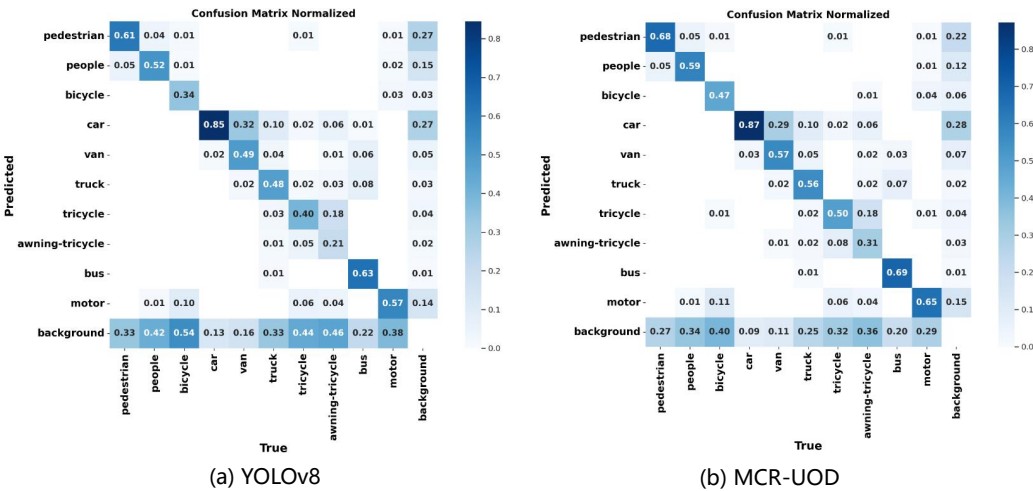

Figure 7: (a) Confusion matrix plot of YOLOv8; (b) confusion matrix plot of our model.

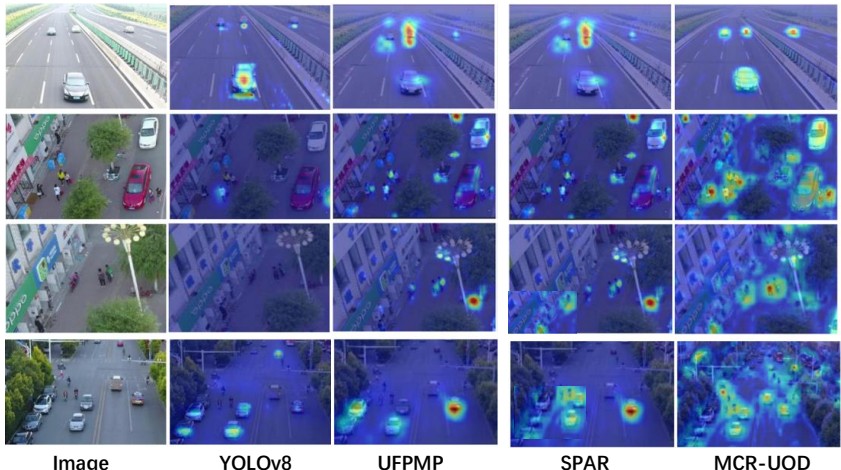

Figure 8: Visualization of feature maps.

a notable advancement in performance over the traditional YOLOv8 model in complex scenarios.

**Visualization of feature maps.** The heatmap visualization of feature maps, shown in Fig. 8, highlights the superior performance of the MCR-UOD method compared to YOLOv8, SPAR [19] and UFPMP [14]. The MCR-UOD heatmaps demonstrate more precise and concentrated activation areas, especially for small objects. This indicates a more refined understanding and localization of critical features in the image. In contrast, UFPMP and SPAR, the previous state-of-the-art methods, while effective, show less focus on these smaller targets. This suggests that MCR-UOD is particularly adapted to capture essential information, leading to enhanced detection and classification performance, especially in scenarios involving small objects.

**Precision-Confidence curve.** Fig. 9(left) presents the Precision-Confidence (PC) curves for the MCR-UOD method, the baseline YOLOv8 and the state-of-the-arts SPAR and UFPMP. The MCR-UOD curve consistently demonstrates high precision across various confidence thresholds, indicating its effectiveness in reducing false positives. In contrast, UFPMP and SPAR exhibit more variability, reflecting less precision stability with changing confidence levels. The smooth and upward trend of the MCR-UOD curve highlights its superior performance and robustness, maintaining a high true positive rate as confidence increases.

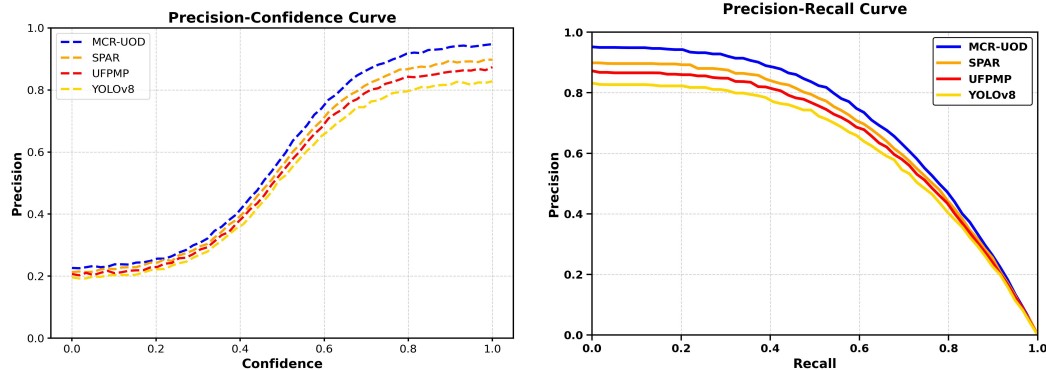

Figure 9: Comparisons of Precision-Confidence and Precision-Recall curves between MCR-UOD and other SOTA methods.

Table 6: Statistical significance (p-values) of performance differences between MCR-UOD and SPAR.

| Metric | SPAR (Mean ± Std) | MCR-UOD (Mean ± Std) | p-value |
|--------|-------------------|----------------------|---------|
| AP50 | 43.90 ± 0.25 | 44.70 ± 0.31 | 0.018 |
| AP75 | 34.70 ± 0.28 | 35.60 ± 0.34 | 0.015 |

This comparison underscores the effectiveness of MCR-UOD in balancing precision and confidence.

**Precision-Recall curve.** Fig. 9(right) presents a comparative analysis of Precision-Recall (PR) curves for the MCR-UOD method, YOLOv8, SPAR and UFPMP. The PR curves clearly illustrate the performance of each model at different recall levels. Our MCR-UOD method consistently demonstrates superior precision compared to YOLOv8, SPAR and UFPMP at various recall rates. This indicates that the MCR-UOD method is more effective in minimizing false positives while maintaining high recall performance. In particular, the PR curve for MCR-UOD is higher than those of other methods in the recall spectrum, reflecting its improved accuracy and robustness in object detection. The area under the PR curve (AP) for MCR-UOD is significantly larger than that of YOLOv8, SPAR and UFPMP, further validating the effectiveness of our method. This improvement in AP underscores MCR-UOD's ability to achieve better precision and recall balance, particularly in detecting small objects and handling imbalanced datasets. Overall, the comparison reveals that MCR-UOD not only surpasses YOLOv8, SPAR and UFPMP in precision but also offers a more reliable detection performance. This indicates that the proposed MCR-UOD method provides substantial enhancement in object detection capabilities, making it more suitable for practical applications where high precision and recall are critical.

**Statistical verification.** To further validate the performance advantage of our proposed MCR-UOD framework, we conducted a statistical significance test against SPAR using the Wilcoxon signed-rank test, as shown in Tabel 6. This test, widely used for paired comparison without assuming data normality, allows us to assess whether the observed improvements are statistically meaningful. We perform the analysis on the UAVDT dataset using two key evaluation metrics: AP50 and AP75. The computed p-values are reported in the corresponding table. Notably, both p-values are well below the 0.1 significance level, providing strong evidence that the performance gains of MCR-UOD over SPAR are not due to random variation. These findings confirm the robustness and consistent superiority of our causal reasoning approach to enhance UAV-based object detection.

## C   Model Architecture with YOLOv8

We implemented the proposed MCR-UOD method based on the YOLOv8 detection framework. The overall architecture is illustrated in Figure 10. YOLOv8 adopts a modern and

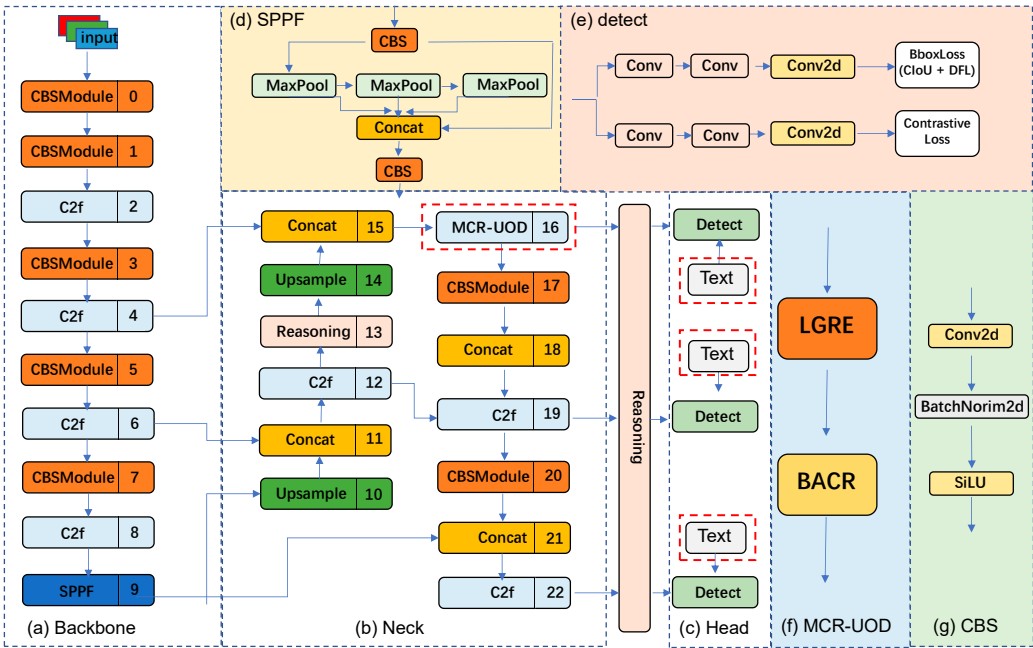

Figure 10: The network structure of YOLOv8 with MCR-UOD. The $w$ (width) and $r$ (ratio) are parameters used to represent the size of the feature map. The size of the model can be controlled by setting the values of $w$ and $r$ to meet the needs of different application scenarios.

streamlined structure composed of a backbone, neck, and detection head, offering improvements in both detection accuracy and speed over previous YOLO versions such as YOLOv5 and YOLOv7.

In our implementation, we retain the original backbone of YOLOv8 and focus on modifying the neck and detection head to incorporate our MCR-UOD strategy. As highlighted in the red box in Figure 10, we replace the last C2f module processing the low-level feature map before the head with a customized version. Specifically, the input feature $C_1$ is passed through two newly designed modules: LGRE and BACR. The output of this process, denoted as $C_1^n$, is then fed into the detection head.

Furthermore, we replace the original classification head with a contrastive head based on text embeddings, as shown in the figure. This change enables the model to perform self-prompted open-set recognition by leveraging text-based semantic information, allowing its potential generalization to unseen object categories.

## D  Limitations

Our method uses multimodal knowledge and causal reasoning to improve object detection on UAV imagery. Although it shows promising results, there are limitations. First, relying on CLIP for semantic guidance limits performance due to its representational capacity, particularly in low-quality or ambiguous images. In addition, prompt design is based on intuition and heuristics, limiting adaptability. Second, the integration of causal reasoning with object detection is still in the early stages. Although we use structural equation models for causal modeling, more research is needed to better link causal structures with image features, especially in complex environments.

