# OpenReview forum: "Multimodal Causal Reasoning for UAV Object Detection"
_NeurIPS.cc/2025/Conference — NeurIPS 2025 poster_

### Official Review · Reviewer_Kr7U · 2025-06-30

**Clarity:** 2
**Significance:** 2
**Originality:** 2
**Rating:** 4
**Confidence:** 2

**Summary:**

This paper presents MCR-UOD, a method for the UAV object detection. The proposed method has two major components: LGRE and BACR. The LGRE part leverages a pretrained text encoder from CLIP model, to compute the attention maps for possible object regions. The BACR part maintains a dynamic confounder dictionary for causal reasoning. Empirical experiments on several benchmarks show the promising results of the proposed method, including performance and efficiency.

**Questions:**

1. One question is - how is the importance of the pretrained CLIP model in proposed method? if changing the CLIP models to a different pretrained models, since two parts of the framework depend on CLIP text encoder.

2. What is the difference between Yolov8 in Table 3 and UAV-YOLOv8 in Table 1?

**Ethical Concerns:**

["NO or VERY MINOR ethics concerns only"]

**Final Justification:**

I read the response and other reviews, thanks for the response, stick to previous score.

**Limitations:**

No limitation section for the submission.

**Paper Formatting Concerns:**

No limitation section for the submission.

**Quality:**

2

**Strengths And Weaknesses:**

Strengths:

The proposed method leverages the pretrained text encoder from the CLIP model, and dynamic confounder dictionary for causal reasoning, demonstrating promising results on UAVDT and VisDrone, and also showing efficiency.


Weaknesses:

Interestingly, it looks Yolov8 is already a very strong baseline for VisDrone, if you directly compare the results from Table 3 with Table 1.  One potential weakness of the paper is the novelty, overall, it is more like an okay, incremental work for the UAV object detection.


(Sorry, I am not familiar with the literature of this topic.)

---

> ### Author Rebuttal · Authors · 2025-07-30
>
> Thank you for your thoughtful review and for recognizing the merits of our work. We have carefully addressed the concerns raised and hope that our answers meet your requirements. We kindly ask you to consider raising the score if possible . Thank you!
>
> **Q1: Interestingly, it looks Yolov8 is already a very strong baseline for VisDrone, if you directly compare the results from Table 3 with Table 1. One potential weakness of the paper is the novelty, overall, it is more like an okay, incremental work for the UAV object detection.**
>
> A: We would like to summarize the significance and novelty of this work again (please also see our responses to reviewer nLTh):
> (1) We agree that YOLOv8 is a strong baseline on the VisDrone dataset, which makes further improvements on top of it particularly challenging. Nevertheless, our method enhances this baseline with causal reasoning and a lightweight architecture. This combination helps to effectively reduce inter-class inconsistency caused by complex scenes and scale variation. As shown in Table 1, our approach achieves significant improvements in both detection accuracy and inference speed.
>
> (2) We would like to take this opportunity to reiterate the key innovations of our work. Unlike incremental improvements that simply build upon existing methods, our approach introduces a novel causality-aware reasoning framework tailored specifically for the unique challenges in UAV object detection, such as inter-class inconsistency caused by complex scenes and scale variations. This fundamental shift in methodology, combined with lightweight design choices, distinguishes our work form prior efforts and leads to significant gains in both accuracy and efficiency. Therefore, we believe our contribution goes beyond a mere incremental step.
>
>
> **Q2: One question is - how is the importance of the pretrained CLIP model in proposed method? if changing the CLIP models to a different pretrained models, since two parts of the framework depend on CLIP text encoder.**
>
> A: The proposed method does not strongly depend on CLIP itself. In our framework, CLIP serves mainly as a vision-language embedding tool to construct the confounder dictionary for causal reasoning. We chose CLIP for its wide adoption and strong generation, but our framework remains compatible with other VLMs, provided they can generate aligned text-image embeddings. We replaced CLIP with OpenCLIP[1] in our experiments, and as shown in Table 1, the results confirm that our method remains effective with alternative VLMs, demonstrating its flexibility and generalizability.
>
> Table 1. Comparison of using CLIP and OPENCLIP.
> |Methods|AP|AP50|AP75|
> |-|-|-|-|
> |YOLOv8|42.2|64.7|44.5|
> |MCR-UOD(CLIP)|44.6|67.3|47.5|
> |MCR-UOD(OPENCLIP)|44.3|67.3|47.2|
>
> **Q3: What is the difference between Yolov8 in Table 3 and UAV-YOLOv8 in Table 1?**
>
> A：Sorry for the confusion. UAV-YOLOv8 [2] is an enhanced version of YOLOv8 specifically tailored for UAV-based object detection. Unlike the original YOLOv8 baseline reported in Table 3, UAV-YOLOv8 (Table 1) integrates three key improvements: a more robust regression loss (WIoU v3), an attention mechanism and a five-scale detection head with FFNB.
>
>
> **Q4: Limitations: No limitation section for the submission.**
>
> A: Thank you for the comment. We would like to clarify that the previous version already contains a detailed discussion of the limitations in the appendix (lines 128–135), which may have been overlooked. We will add a concise summary of this section to the main text for better visibility.
>
>
> **References**
>
> [1] Cherti M,et al. “Reproducible scaling laws for contrastive language-image learning”,CVPR2023.
>
> [2] Wang G, et al. “UAV-YOLOv8: A small-object-detection model based on improved YOLOv8 for UAV aerial photography scenarios”. Sensors, 2023.

---

> > ### Author Response · Authors · 2025-08-06
> >
> > Dear Reviewer Kr7U,
> >
> > Thanks again for the valuable comments and suggestions. Since some time has passed since the rebuttal began, we wondered if the reviewer might still have any concerns that we could address. We believe our point-by-point responses addressed all the questions/concerns.
> >
> > It would be great if the reviewer could kindly check our responses and provide feedback with further questions/concerns (if any). We would be more than happy to address them. Thank you!
> >
> > Best regards, Paper 10728 Authors

---

> ### Comment · Area_Chair_U5ww · 2025-08-05
> **Update after rebuttal**
>
> Dear Reviewer,
>
> The authors’ rebuttal has been posted. Please check the authors’ feedback, evaluate how it addresses the concerns you raised, and post any follow-up questions to engage the authors in a discussion. Please do this ASAP.
>
> Thanks.
>
> AC

---

> ### Comment · Area_Chair_U5ww · 2025-08-08
> **Response to authors' rebuttal**
>
> Dear Reviewer,
>
> Could you please look at the authors' rebuttal and comments from other reviewers and make your final rating with justifications? Please do this asap as the deadline is approaching.
>
> Thank you.
>
> AC

---

### Official Review · Reviewer_Wk5t · 2025-07-01

**Clarity:** 3
**Significance:** 3
**Originality:** 3
**Rating:** 4
**Confidence:** 4

**Summary:**

This paper introduces multimodal Causal Reasoning framework based on YOLO backbone for UAV Object Detection (MCR-UOD). The approach to utilizes a vision-language model as part of the detection model to exploit multimodal knowledge and better understand and correlate visual and textual information.

**Questions:**

- In page 7 lines 285-286 the initial learning rate is reported as 0.001 and the final is 0.01. Usually, the learning rate reduced over the duration of the training. Is this a typo?
- With reference to Table 4. Why does the parameters of the MCR-UOD models are lower than the original models? Don’t the language model parameters increase the total count?

**Ethical Concerns:**

["NO or VERY MINOR ethics concerns only"]

**Final Justification:**

The authors have included additional experiments with related work and have demonstrated better their motivation for this work. For this reason I have raised my score.

**Limitations:**

No discussion of limitations in the main manuscript but in supplementary. Perhaps a discussion on the impact of biases introduced because of the language model could be useful.

**Paper Formatting Concerns:**

No major formatting issues spotted.

**Quality:**

3

**Strengths And Weaknesses:**

Strengths:

- The language guided approach to enhance object localization is interesting and effective in improving the average precision metrics.
- Common datasets are used for evaluation such as UAVDT, HRSC2016, and VisDrone.
- Comparisons are also made with a number of established methods.
- The presented approach with the incorporation of CLIP text encoder seems to be easily adaptable to other settings.

Weaknesses:

- The incorporation of CLIP with detection models is not new and has been demonstrated in other works.
- The connection with UAV-based visual data is weak and in this regard it is not clear why the particular application domain was selected.
- While the comparisons with vision models are adequate, the paper is missing comparisons with text based YOLO models such as (Yolo-world CVPR, YOLOE).

---

> ### Author Rebuttal · Authors · 2025-07-30
>
> Thank you for your constructive feedback and for acknowledging the strengths of our work. We have carefully addressed the concerns raised and sincerely hope our responses clarify the contributions and improvements made. We would greatly appreciate your consideration of a higher score if you find the revisions satisfactory.
>
> **Q1: The incorporation of CLIP with detection models is not new and has been demonstrated in other works.**
>
> A: We would like to clarify our contribution is not a simple fusion of CLIP and detection models. (1) CLIP usage is different. Unlike previous works that focus on integrating CLIP directly into detection models, our approach leverages CLIP more as an auxiliary tool to enable causal reasoning in challenging UAV scenarios. Our core idea is to utilize a vision-language model to generate embeddings of diverse textual descriptions, which are then used to construct a confounder dictionary. This dictionary facilitates causal reasoning to address inter-class inconsistencies caused by complex scenes and scale variations in UAV data. So our method is independent to CLIP at inference stage. Of course, other VLMs can also be used instead of just CLIP.
>
> (2) Our innovation lies in introducing a causality-aware framework that goes beyond fusion of CLIP. By embedding language semantics into LGRE and and causal reasoning(BACR), the method reduces visual ambiguity and improves generation, especially for small or confusing UAV objects. This presents a novel and effective use of VLMs beyond conventional fusion-based approaches.
>
>
> **Q2: The connection with UAV-based visual data is weak and in this regard it is not clear why the particular application domain was selected.**
>
> A: We appreciate the reviewer’s comment and would like to clarify the motivation behind choosing the UAV-based visual domain. As mentioned in lines 24-32 on the first page, UAV scenarios are characterized by complex scenes, frequent scale variations, and a high degree of inter-class visual ambiguity, which are challenges that often degrade detection performance. Our framework is specifically designed to address these issues through causality-aware reasoning and enhanced semantic alignment. Therefore, UAV-based detection is not only a representative but also a highly relevant application domain for validating the effectiveness of our method. Experimental results on VisDrone and UAVDT further support this choice.
>
> **Q3: While the comparisons with vision models are adequate, the paper is missing comparisons with text based YOLO models such as (Yolo-world CVPR, YOLOE).**
>
> A: Thank you for your valuable comments. We have added comparison experiments with YOLO-World[1] and YOLOE[2].The results demonstrate that our method still achieves superior detection performance compared to other text-based YOLO detectors.
>
> Table 1. Comparison of YOLO-World and YOLOE.
> |Methods|AP|AP50|AP75|
> |-|-|-|-|
> |YOLOv8|42.2|64.7|44.5|
> |YOLO-World|42.8|65.0|44.9|
> |YOLOE|43.1|65.4|45.3|
> |**MCR-UOD (Ours)**|**44.6**|**67.3**|**47.5**|
>
> **Q4: In page 7 lines 285-286 the initial learning rate is reported as 0.001 and the final is 0.01. Usually, the learning rate reduced over the duration of the training. Is this a typo?**
>
> A: We apologize for the confusion caused by this typo, we will correct it in the paper. In our experimental settings, we set lr0 (the initial learning rate) to 0.001 and lr1 to 0.01, resulting in a final learning rate of lr0*lr1.
>
> **Q5: With reference to Table 4. Why does the parameters of the MCR-UOD models are lower than the original models? Don’t the language model parameters increase the total count?**
>
> A: Thank you for your valuable comments. As stated on page 8, line 329 (Main text), and page 6 (Appendix), line14. The reduction in model parameters primarily stems from compression both visual and text features to a unified channel dimension (256), whereas YOLOv8 uses higher dimensions (1024 and 512) in its detection head. This adjustment leads to a more lightweight structure. Importantly, despite the lower parameter count, MCR-UOD maintains competitive detection performance, effectively meeting the demands of UAV-based detection tasks. We will provide additional details in the relevant sections.
>
> **Q6: Limitations: Perhaps a discussion on the impact of biases introduced because of the language model could be useful.**
>
> A: Thank you for the suggestion. We agree that discussing potential biases from the language model is important and will add this in the revised version. Such bias often stem from imbalanced or incomplete training data, which may affect semantic guidance and lead to skewed reasoning in some scenarios. However, systematically analyzing these bias is challenging, as it requires controlled experiments, access to training data distributions, and clear fairness definitions across tasks and categories. While these bias exists, large scale dataset pretraining allows the model to capture useful semantics, particularly for common UAV categories like vehicles and pedestrians, where it has helped improve detection.
>
> &nbsp;
>
> **References**
>
> [1] Cheng T, et al. “Yolo-world...”.CVPR2024.
>
> [2] Wang A, et al. “Yoloe...”. ICCV2025.

---

> > ### Author Response · Authors · 2025-08-06
> >
> > Dear Reviewer Wk5t,
> >
> > Thanks again for the valuable comments and suggestions. Since some time has passed since the rebuttal began, we wondered if the reviewer might still have any concerns that we could address. We believe our point-by-point responses addressed all the questions/concerns.
> >
> > It would be great if the reviewer could kindly check our responses and provide feedback with further questions/concerns (if any). We would be more than happy to address them. Thank you!
> >
> > Best regards, Paper 10728 Authors

---

> > > ### Comment · Reviewer_Wk5t · 2025-08-06
> > >
> > > Thank you for your hard work and detailed response. I appreciate the effort you put into addressing my concerns. Please see below some comments based on the response:
> > >
> > > - Still I am not convinced with the choice of UAV applications and not any other autonomous vehicle or robotics application that also have the characteristics mentioned.
> > >
> > > - Thank you for adding the YOLOE and YOLO-World comparisons. This further shows that the approach demonstrated improved performance.
> > >
> > > - I think I did not receive a direct answer to the question "Don’t the language model parameters increase the total count?" in other words do the parameter counts include those of CLIP?

---

> > > > ### Author Response · Authors · 2025-08-06
> > > >
> > > > Dear Reviewer Wk5t,
> > > >
> > > > We sincerely thank you for your valuable feedback and constructive suggestions. Below, we provide point-by-point responses to each of your comments. If you have any further questions or concerns, we would be happy to address them.
> > > >
> > > > **Q1: Still I am not convinced with the choice of UAV applications and not any other autonomous vehicle or robotics application that also have the characteristics mentioned.**
> > > >
> > > > A: Sorry for confusion. Although other autonomous driving or robotics applications share some common characteristics, UAVs present unique challenges:
> > > >
> > > > (1) In the VisDrone dataset, object sizes vary dramatically—from about 0.01% up to several tens of percent of the image area—showing more extreme scale variation than datasets like COCO.
> > > >
> > > > (2)UAV scenes contain diverse backgrounds such as urban buildings, roads, greenery, and water bodies, which frequently change with flight altitude and viewpoint, resulting in greater visual variability.
> > > >
> > > > Therefore, our method is specifically designed to address these UAV-specific challenges of extreme scale variation and increased visual variability. It integrates causal reasoning with VLM-based semantic priors, effectively leveraging the causal structure inherent to UAV scenarios to reduce visual ambiguity and improve detection accuracy. By focusing on UAV applications first, we can fully demonstrate the potential of our approach and lay a solid foundation for extensions to autonomous driving, robotics, and other fields.
> > > >
> > > > **Q2：Thank you for adding the YOLOE and YOLO-World comparisons. This further shows that the approach demonstrated improved performance.**
> > > >
> > > > A: Thank you very much for your positive feedback and recognition. We appreciate your support.
> > > >
> > > > **Q3：I think I did not receive a direct answer to the question "Don’t the language model parameters increase the total count?" in other words do the parameter counts include those of CLIP?**
> > > >
> > > > A: Thank you for your question and we apologize for the oversight in not making this point clearer earlier. The parameters of the language model are not included in the total parameter count because it is only used during training as an auxiliary tool and not part of the inference model. We have counted only the parameters of the inference model, which is a common and reasonable practice.

---

> > > > > ### Comment · Reviewer_Wk5t · 2025-08-07
> > > > >
> > > > > Thank you for the clarifications. Based on the revisions I have decided to raise my score.

---

> > > > > > ### Author Response · Authors · 2025-08-07
> > > > > >
> > > > > > Dear Reviewer Wk5t,
> > > > > >
> > > > > > We sincerely appreciate your time and effort in reviewing our response. Your insightful suggestions have played an important role in improving the overall quality of our paper. We are committed to ensuring that the revised manuscript fully addresses the points you raised. We truly appreciate your thoughtful feedback once again.
> > > > > >
> > > > > > Best regards, Paper 10728 Authors

---

> ### Comment · Area_Chair_U5ww · 2025-08-05
> **Update after rebuttal**
>
> Dear Reviewer,
>
> The authors’ rebuttal has been posted. Please check the authors’ feedback, evaluate how it addresses the concerns you raised, and post any follow-up questions to engage the authors in a discussion. Please do this ASAP.
>
> Thanks.
>
> AC

---

### Official Review · Reviewer_dsXu · 2025-07-01

**Clarity:** 4
**Significance:** 3
**Originality:** 3
**Rating:** 4
**Confidence:** 4

**Summary:**

The paper introduces MCR-UOD, a UAV object detection framework using causal reasoning and multimodal learning. It includes LGRE for text-guided object region highlighting and BACR for robust feature extraction via causal interventions. MCR-UOD achieves state-of-the-art performance on VisDrone and UAVDT datasets while maintaining computational efficiency.

**Questions:**

1. Ablation Study: Does the Backdoor Adjustment Causal Reasoning (BACR) still improve performance if the top-N selection mechanism is removed?
2. Ablation Study: If the mean CLIP feature is not updated, does BACR still enhance performance?
3. What is the source of performance improvement—classification accuracy or more precise bounding box regression? What specific problem does the proposed method solve?
4. The YOLOv8 baseline in the paper uses category text embeddings. How does it compare to a learnable classifier (as in traditional detectors), and why were category text embeddings chosen?
5. Is there any other CLIP enhance detector could chose as a more fair baseline?

**Ethical Concerns:**

["NO or VERY MINOR ethics concerns only"]

**Final Justification:**

The authors' response has addressed most of my concerns. Overall, this is a rather interesting piece of work. I decide to keep my score. I did not further increase the score because, in my view, the most common approach to handling different scenarios in practical applications is to augment data related to specific aspects. The method of extracting features using language models, which requires alignment in the feature space, generally yields inferior performance compared to the former. This limits the deployment of this work in real-world scenarios. However, the idea behind this work is still quite interesting, so I have kept the score unchanged overall.

**Limitations:**

No. The authors should discuss whether the work and its application methods remain generalizable to other tasks or categories.

**Quality:**

3

**Strengths And Weaknesses:**

Strengths:
1. The writing is polished and lucid, ensuring high readability.
2. Proposes a feature enhancement module: Language-Guided Region Exploration (LGRE), which leverages CLIP text features of given classes to filter feature maps, making them more sensitive to object presence.
3. Introduces Backdoor Adjustment Causal Reasoning (BACR): Constructs a mean CLIP text feature for the target class across diverse conditional texts (e.g., different weather conditions), enhances image feature maps via cross-attention, and updates the mean feature using a running mean method in conjunction with image features.
4. The method achieves state-of-the-art (SOTA) performance, demonstrating significant improvements over the baseline.
5. The ablation study is relatively clear.

Weaknesses:
1. Ablation Study: Does the Backdoor Adjustment Causal Reasoning (BACR) still improve performance if the top-N selection mechanism is removed?
2. Ablation Study: If the mean CLIP feature is not updated, does BACR still enhance performance?
3. What is the source of performance improvement—classification accuracy or more precise bounding box regression? What specific problem does the proposed method solve?
4. The YOLOv8 baseline in the paper uses category text embeddings. How does it compare to a learnable classifier (as in traditional detectors), and why were category text embeddings chosen?
5. Is there any other CLIP enhance detector could chose as a more fair baseline?

---

> ### Author Rebuttal · Authors · 2025-07-30
>
> We thank the reviewer for the very encouraging comments on the originality of MCR-UOD, the effectiveness of MCR-UOD, and the overall good presentation. We hope to provide satisfying answers to the concerns raised. We kindly ask you to consider raising the score if possible . Thank you!
>
> **Q1: Ablation Study of top-N selection mechanism.**
>
> A: We added ablation experiments on VisDrone to assess the top-N selection mechanism, as shown in Table 1. Even without it, the model still improves performance.
>
> Table 1. Ablation study on whether using top-N selection mechanism.
> | Methods|AP|AP50|AP75|
> |:-:|:-:|:-:|:-:|
> |YOLOv8|42.2|64.7|44.5|
> |w/o Top-N|43.8|66.6|46.4|
> |**Top-N selection**|**44.6**|**67.3**|**47.5**|
>
>
> **Q2: Ablation Study on whether updating mean CLIP feature.**
>
> A: We added an ablation study on whether to update the CLIP features. As shown in Table 2, experimental results show that without updating the mean CLIP features, BACR can still improve performance. However, the improvement is slightly weaker compared to the setting where the feature dictionary is updated.
>
> Table 2. Ablation study on whether updating mean CLIP feature.
> |Methods|AP|AP50|AP75|
> |:-:|:-:|:-:|:-:|
> |YOLOv8|42.2|64.7|44.5|
> |w/o Updated|44.2|66.8|46.8|
> |**Updated Feature**|**44.6**|**67.3**|**47.5**|
>
>
> **Q3: What is the source of performance improvement—classification accuracy or more precise bounding box regression? What specific problem does the proposed method solve?**
>
> A: We conducted a more detailed analysis on experimental results which confirm that both classification accuracy and bounding box regression contribute to the performance gain. As shown in Table 3, the reduction in cls_loss indicates improved classification accuracy, while the decreases in box_loss and dfl_loss reflect more precise bounding box regression. These are further supported by the higher precision and recall, showing an overall enhancement in detection quality.
>
> Table 3. Analysis on performance improvement.
> |Methods|box_loss|dfl_loss|cls_loss|precision|recall|
> |:-:|:-:|:-:|:-:|:-:|:-:|
> |YOLOv8|1.14967|1.03289|0.61914|0.67492|0.61999|
> |MCR-UOD|1.07320|1.01879|0.60486|0.69434|0.63125|
>
>
> **Q4: The YOLOv8 baseline in the paper uses category text embeddings. How does it compare to a learnable classifier (as in traditional detectors), and why were category text embeddings chosen?**
>
> A: (1) We compared the baseline using a learnable classifier with our method based on CLIP classification. As shown in Table 4, using CLIP text embeddings for classification yields a noticeable performance improvement over the learnable classifier.
>
> Table 4. Performance comparison of learnable classifier with CLIP text embeddings.
> |Methods|AP|AP50|AP75|
> |:-:|:-:|:-:|:-:|
> |YOLOv8(learnable classifier)|41.3|64.2|43.1|
> |YOLOv8(CLIP)|42.2|64.7|44.5|
>
> (2) We chose CLIP text embeddings because text and image features of the same class are tightly clustered, while those of different classes are clearly separated as CLIP model is pretrained on millions of text-image pairs. By employing these embeddings in both the LGRE and BACR modules and optimizing classification loss end-to-end, the model better integrates semantic information and enhances feature consistency.
>
> **Q5: Is there any other CLIP enhance detector could chose as a more fair baseline?**
>
> A: We have added comparison results with YOLO-World[1] and YOLOE[2], which are also CLIP enhanced YOLO detectors. As shown in Table 5, our method achieves better detection performance compared to other CLIP enhanced detectors.
>
> Table 5.Comparison of different CLIP enhanced detectors.
> |Methods|AP|AP50|AP75|
> |:-:|:-:|:-:|:-:|
> |YOLOv8|42.2|64.7|44.5|
> |YOLO-World|42.8|65.0|44.9|
> |YOLOE|43.1|65.4|45.3|
> |**MCR-UOD (ours)**|**44.6**|**67.3**|**47.5**|
>
> **Q6: Limitations:No. The authors should discuss whether the work and its application methods remain generalizable to other tasks or categories.**
>
> A: Thank you for the suggestion. We would like to clarify that the limitations of our method are discussed in the appendix (lines 128–135). Regarding generalizability, although our framework is developed for UAV detection, its core components, such as multimodal semantic guidance and causal reasoning, are not restricted to this task. Theoretically, our method can also be applied to other scenarios, including transfer learning and zero-shot detection. We consider moving this discussion to the main text for easier reader access.
>
> **References**
>
> [1]Cheng T, et al. “Yolo-world...”.CVPR2024
>
> [2]Wang A, et al. “Yoloe...”. ICCV2025

---

> > ### Author Response · Authors · 2025-08-06
> >
> > Dear Reviewer dsXu,
> >
> > Thanks again for the valuable comments and suggestions. Since some time has passed since the rebuttal began, we wondered if the reviewer might still have any concerns that we could address. We believe our point-by-point responses addressed all the questions/concerns.
> >
> > It would be great if the reviewer could kindly check our responses and provide feedback with further questions/concerns (if any). We would be more than happy to address them. Thank you!
> >
> > Best regards, Paper 10728 Authors

---

> ### Comment · Area_Chair_U5ww · 2025-08-05
> **Update after rebuttal**
>
> Dear Reviewer,
>
> The authors’ rebuttal has been posted. Please check the authors’ feedback, evaluate how it addresses the concerns you raised, and post any follow-up questions to engage the authors in a discussion. Please do this ASAP.
>
> Thanks.
>
> AC

---

> ### Comment · Reviewer_dsXu · 2025-08-07
>
> The authors' response has addressed most of my concerns. Overall, this is a rather interesting piece of work. I decide to keep my score.
> I did not further increase the score because, in my view, the most common approach to handling different scenarios in practical applications is to augment data related to specific aspects. The method of extracting features using language models, which requires alignment in the feature space, generally yields inferior performance compared to the former. This limits the deployment of this work in real-world scenarios. However, the idea behind this work is still quite interesting, so I have kept the score unchanged overall.

---

> > ### Author Response · Authors · 2025-08-07
> >
> > Dear Reviewer dsXu,
> >
> > Thank you for your thoughtful feedback and for recognizing the value of our work. We understand that data augmentation, especially when tailored to specific domains, is a widely used and effective strategy in practical applications. We would like to emphasize that such approaches are not in conflict with our method. In fact, our language-model-based feature extraction can complement domain-specific augmentation by introducing high-level semantic understanding. When combined, they have the potential to further enhance overall performance. We truly appreciate your constructive comments and continued consideration.
> >
> > Best regards, Paper 10728 Authors

---

### Comment · Area_Chair_U5ww · 2025-08-01
**Authors' rebuttal posted and discussion**

Dear Reviewers,

Thank you for your efforts in reviewing this paper. The authors' rebuttal has been posted. This paper received diverse initial ratings. Please read the rebuttal materials and comments from other reviewers to justify if your concerns have been resolved and update your final rating with justifications.

AC

---

### Decision · Program_Chairs · 2025-09-17

**Decision:**

Accept (poster)

**Comment:**

This paper investigates multimodal casual reasoning for UAV object detection. Initially, some reviewers have questions regarding the motivation and experimental analysis of this work. The authors provide a good rebuttal addressing most of the concerns and questions. After rebuttal, all reviewers are positive on this work. There is no reason to overturn the reviewers’ comment. The AC recommends accepting this work. The authors should interest their response in rebuttal in the final revision.